# The Role of Research in Guiding Treatment for Women’s Health: A Qualitative Study of Traditional Chinese Medicine Acupuncturists

**DOI:** 10.3390/ijerph18020834

**Published:** 2021-01-19

**Authors:** Mike Armour, Debra Betts, Kate Roberts, Susanne Armour, Caroline A. Smith

**Affiliations:** NICM Health Research Institute, Western Sydney University, Sydney 2517, Australia; debra.betts@rhizome.net.nz (D.B.); katerobertsnz@gmail.com (K.R.); S.Armour@westernsydney.edu.au (S.A.); caroline.smith@westernsydney.edu.au (C.A.S.)

**Keywords:** acupuncture, Chinese medicine, women’s health, case reports, clinical trials

## Abstract

Background: Surveys of acupuncture practitioners worldwide have shown an increase in the use of acupuncture to treat women’s health conditions over the last ten years. Published studies have explored the effectiveness of acupuncture for various conditions such as period pain, fertility, and labor induction. However, it is unclear what role, if any, peer-reviewed research plays in guiding practice. Methods: Acupuncturists with a significant women’s health caseload were interviewed online in three small groups to explore factors that contribute to acupuncturists’ clinical decision made around treatment approaches and research. Results: Eleven practitioners participated in the focus groups. The overarching theme that emerged was one of ‘Not mainstream but a stream.’ This captured two themes relating to acupuncture as a distinct practice: ‘working with what you’ve got’ as well as ‘finding the right lens’, illustrating practitioners’ perception of research needing to be more relevant to clinical practice. Conclusions: Acupuncture practitioners treating women’s health conditions reported a disconnect between their clinical practice and the design of clinical trials, predominantly due to what they perceived as a lack of individualization of treatment. Case histories were popular as a learning tool and could be used to support increasing research literacy.

## 1. Introduction

Women in Australia often report using complementary therapies such as acupuncture to manage symptoms of period or pelvic pain [1], for fertility [2,3], and for pregnancy-related conditions such as back pain [4,5]. Our team previously undertook a survey of how acupuncturists in Australia and New Zealand treat women’s health conditions in 2013 [6], and found that women were presenting for treatment to acupuncturists for a number of conditions across their reproductive lifespan including menstrual disorders, fertility, and pregnancy-related conditions.

Surveys of acupuncture practitioners in the United Kingdom (UK), European Union (EU), and China support these Australian findings and show a consistent increase in the number of women presenting for women’s health conditions [7,8]. This increasing popularity has meant a growth in practitioners’ caseloads, specialization, and interest in research about women’s health [7,8].

Since our original survey, there have been a number of studies demonstrating effectiveness of acupuncture in women’s health conditions, including period pain [9,10], premenstrual syndrome (PMS) [11], and antenatal depression [12]. However, research has also been published that has demonstrated a lack of efficacy for other conditions such as an adjunct to in-vitro fertilization (IVF) [13], or for inducing labor [14]. In addition, some research has shown that acupuncture may only be beneficial under specific circumstances [13] and that dosage components, such as the number of treatments given, might be an important factor in therapeutic outcomes [9,10,13].

Previous research has shown that clinical trial results do not always change practice for acupuncture practitioners [15,16] and that acupuncturists often struggle to be able to get women to come in as frequently as might be required to replicate clinical trial results [17]. It is also unclear if negative or equivocal results change practice, with recent data from the UK showing that despite the publication of a Cochrane review in 2017 reporting acupuncture as ineffective for labor induction [14], this is still commonly practiced [18].

The aim of these online interviews was to explore what influences acupuncturists’ decisions around treatment planning (e.g., research, personal experience, expert opinion), how they incorporate this into their clinical practice, and the barriers or facilitators to incorporating research results into practice.

## 2. Methodology

A qualitative approach using online small group interviews was seen as appropriate to seek out a deeper understanding of acupuncturists’ experiences and decision making. These interviews were part of a larger mixed methods study which included an international survey on how acupuncturists in Australia and New Zealand treated women’s health conditions. Survey participants who expressed a willingness to participate in the group interviews were sent an invitation to participate with information about how their data would be used and any comments used in the interviews deidentified. They were also given several dates for the interviews in January and February 2020 so they could indicate their availability along with a consent from to return. Ethical approval was obtained from Western Sydney University Human Ethics Committee (H13099—approved 5 February 2019). Small group interviews were chosen due to the barriers of getting larger numbers of practitioners together at one time due to clinical commitments.

The study population was drawn from the international survey (not yet published but whose content was similar to our previously published survey [6]), which had inclusion criteria of being a New Zealand and Australian acupuncturist, who were aged over 18, either a current member of Acupuncture NZ or New Zealand Acupuncture Standards Authority (NZASA) (if located in New Zealand) or hold current Chinese Medicine Practitioner registration with the Australian Health Practitioner Regulation Agency (AHPRA) (if located in Australia). All small group interview participants were required to have been in clinical practice at least one day per week for the past 12 months and with a case load of at least 1/3 women’s health-related conditions. Purposive sampling was used based on the assumption that those with significant experience in women’s health would yield significant information and provide unique perspectives based on their experience in the field [19] and in line with generally accepted approaches for qualitative descriptive analysis [20].

### 2.1. Positioning of the Researcher

All of the researchers have formal acupuncture qualifications, with three of these (M.A., D.B., K.R.) remaining involved in providing acupuncture in a clinical practice setting. With previous publications involving acupuncture use for women’s health (M.A., D.B., and C.S.), it is important to recognize as researchers that this confers an ‘insiders perspective’ and specific interest in designing and implementing research to explore how research translates into clinical practice for acupuncturists.

### 2.2. Group Interviews

The interviews which were between 60 and 75 min and were recorded online using the Zoom platform with a single researcher (M.A. or S.A.) leading the interview. The audio recordings were then professionally transcribed. The interview guide can be found in Appendix A. At the beginning of the interview, participants were asked to briefly introduce themselves to the group with their name, where they practiced, and how many years they had been in practice. They were then asked how women were referred to their clinic, how they communicated with other healthcare providers, what they saw as barriers to treatment, and the relevance of research to clinical practice, with specific questions such as: what do you do to keep your knowledge of women’s health up to date? How valuable are academic articles such as randomized controlled trials for your practice? Do they influence your practice at all? Is there anything that would help make research results easier to translate into your clinical practice?

Transcripts of interview data were not returned to the participants. It has been reported that this results in interviewees revising transcripts in such a way that no longer accurately reflects the interaction and that it is likely any potential benefits of producing higher quality transcripts is not justified [21]. To ensure anonymity but to allow identification of different participants through various themes, each quotation indicated which interview group was used (IG) and which participant in that group was speaking (P), e.g., Participant 3 in Interview Group 1 would be represented as IG 1 P 3.

### 2.3. Data Analysis

A qualitative descriptive approach was used to formally analyze and code the data [22]. Qualitative description research seeks instead to provide a rich description of the experience depicted in easily understood language [23] and seeks to discover and understand a phenomenon, a process, or the perspectives and worldviews of the people involved [24]. A thematic analysis was undertaken as per Boyatzis [25]. Initial coding was carried out by one researcher (S.A.) and checked, and after feedback and discussion, verified by D.B. Dominant and/or repeated codes were then categorized into themes along with quotations summarizing specific themes. After a consensus was reached regarding classification of codes and themes, quotations in each domain were summarized and presented in a qualitative descriptive manner. To ensure anonymity but to allow identification of different participants through various themes, each quotation indicated which interview group was used (IG) and which participant in that group was speaking (P), e.g. Participant 3 in Interview Group 1 would be represented as IG 1 P 3.

## 3. Results

A total of 80 participants left their contact details, indicating that they meet the interview criteria and were interested in participating. Of these, 48 were from Australia (AUS) and 32 were from New Zealand (NZ). Fifteen returned the consent forms and agreed to participate, with eleven participants finally taking part on the day—six from Australia and five from New Zealand. Of the four who signed consent forms and did not participate, all had last-minute clinical commitments which prevented participation and could not make an alternate session. Three small interview groups were held with three to four participants in each group. Three practitioners were in a combined Australian and New Zealand group, four in an Australian group, and four in a New Zealand group. All were members of their respective professional bodies, all held at least a Bachelors or equivalent level degree in acupuncture, and ten of the eleven participants had done their undergraduate training in their respective country of current residence, with one participant having trained in China. Table 1 outlines the demographics of each interview group.

### 3.1. Findings

The overarching theme that emerged was one of ‘Not mainstream but a stream.’ This captured the themes that related to acupuncturists’ experience of practice and ‘working with what you’ve got’, as well as a theme illustrating practitioners’ perception of research, ‘finding the right lens.’ Table 2 outlines the themes and subthemes.

### 3.2. Overview of Themes

‘Not mainstream but a stream’ was a central unifying theme capturing how practitioners saw themselves practicing in a way that remained differentiated from mainstream Western medical healthcare: “We’re a whole lot closer than we were 20 years ago. But no, we’re not mainstream yet.” Interview group 1, participant 3 (IG 1 P 3). This influenced their clinical practice and their postgraduate education as well as their expectations of research and the relevance it had to their clinical practice. These findings are captured in the following themes and subthemes.

### 3.3. Working with What You’ve Got

A pragmatic approach was utilized for treatment delivery decisions and post-graduate education. Practitioners discussed how individual diagnosis was a fundamental cornerstone of clinical practice. However, all groups spoke of needing to adapt ‘ideal’ treatment and how being unable to treat as frequently as they would like influenced the results seen in clinical practice.

#### 3.3.1. “Pragmatic Treatment”

Practitioners saw themselves making individual diagnoses and adjusting treatment plans to meet patients’ financial, time, and geographical constraints.


*Sometimes you don’t have a lot of time, sometimes they come in, and they’re having an IVF cycle in two weeks’ time. So, you work with what you’ve got. And in an ideal world, I want to work with them a lot longer than that.*
IG 2 P4

*The other thing that I have a little bit of difficulty with is geographic isolation. I treat people from* [place name] ….. *So, it’s a big catchment. So, women can’t come every week. They certainly can’t come two or three times a week. So, we just have to work with it.*IG 1 P 3

However, despite these constraints, practitioners spoke of the commitments they made to their patients. These included being available at key times, offering payment options such as low-cost clinics, and offering herbs, diet, and lifestyle advice that could be used between treatments.


*I would prefer, like in China, to do five days a week. But when they can only come once a week, or if they’re doing both the woman’s health clinics twice a week at the reduced cost, I’ll still do things like ear points or give them lifestyle advice or herbal medicine so that they’re getting the daily effects of treatment not just once or twice a week.*
IG 2 P 2


*Especially with the fertility patients, you’re making quite a big commitment as a practitioner. I will go in on a Sunday, I’ve even gone in on a New Year’s Day.*
IG 2 P 4

#### 3.3.2. “Working It Out for Yourself”

Practitioners also had a pragmatic approach to post-graduate education, discussing ongoing education as an informal process from their personal experiences and interests. Clinical knowledge was built through life experiences, studying texts, and webinars they selected for specific interest.


*I suppose over the long term, this is what works, just doing specific seminars or different books, so you learn overtime what works best.*
IG 3 P 4


*I actually remember when I was at school being told don’t go near pregnant women…So, I pretty much worked out myself, read everything that I could find, which wasn’t a great deal.*
IG 2 P 3

Several practitioners also spoke of how they reached out to their colleagues. This was usually referred to as an informal process, although one practitioner spoke of a specific group that came together regularly with the specific aim of education.


*We’ve got a peer support group going, it’s specifically around women’s health issues that we meet every three months. And at that, we look at what’s new in research. We might discuss a webinar, but it’s all based around educational content. And then aside from those meetings, that also provides a network of practitioners to discuss things. So, we have a Facebook page and we just pick up the phone and call someone, or we might put out to the whole group, “Look, I’ve had this and has anyone else had any experience with this? What do you think about this?” And get this conversation going. I find it really, really helpful.*
IG 2 P 4

### 3.4. Finding the Right Lens

Practitioners reported that research as they perceived it did not reflect their clinical practice. “Overlaying the Western science mind on to Chinese medicine, it just doesn’t fit… It’s utterly the wrong lens” (IG 1 P 4), with subthemes expressing how research needed to come from clinical practice to be relevant, and that current research was more useful to inform medical practitioners and patients. Those interested in learning more about the research base thought it would be more accessible to them if it were translated into a ‘digestible’ format and delivered by practitioners they could trust.

#### 3.4.1. “Adapting Research to a Clinic Lens”

Practitioners discussed that to be meaningful, research needed to be adapted to reflect an individualized approach and capture their clinical experience, suggesting this could be through case histories and collating clinical data.


*One of the major issues with acupuncture research is that the research can’t be individualized to each case.*
IG 3 P 5


*Unfortunately, a lot of our research people are one step away from clinical practice and clinical practice is an enormous resource that I think we underuse. I think we could be doing a lot more with sharing what we actually do in practice. It’s quite different from what they do in a clinical trial.*
IG 1 P3

Practitioners also expressed how beneficial it would be to see types of research comparing different aspects of clinical practice to aid their practice decisions.


*If we can obtain with a bit more evidence, say, you do need to come in three times a week, because this bit of research showed that we’ve got far better results than once a week, do you know what I mean? And maybe you’ll alter your costing to make that happen… this condition actually electro-acupuncture is better than the manual—let’s compare some things, what actually does work better or more optimally.*
IG2 P4

#### 3.4.2. ‘More for the Medics and Patients’

There was also a perception that the benefits of current research were really for “others”—those outside the acupuncture or more broad allied health community.


*I personally don’t utilize research in my practice, I see more value in research, evidence-based research, for the medics and for the public to see the value of what we are doing.*
IG 1 P 2

There was also a view expressed by several practitioners that they need to educate the public about how the research they might come across was vastly different to the way they practiced.


*I generally end up explaining why I’m not going to do what the research says, describing individualized treatment and how it’s far more important and how you actually get better results with individualized treatment than you do by using a straight-out protocol.*
IG1 P3

#### 3.4.3. ‘Making Research Digestible’

Practitioners expressed how research would be easier to understand if presented in different formats such as summaries and discussions that related to clinical practice.


*Presenting that research to you in a way that you can digest it and in a way that’s really relevant to the patients you see in your clinic, whenever that happens that’s when I’m most interested and most likely to apply that research to my actual clinical practice.*
IG 3 P 5


*It is sometimes hard to understand is this a good piece of research or not? […] So, I think summaries in a newsletter would be—I would personally find useful.*
IG 2 P 4

#### 3.4.4. ‘Teachers You Can Trust’

Practitioners also discussed how they were more engaged when research was presented from someone they trusted, such as someone practicing within a similar acupuncture field that they could identify with.


*I don’t mind reading academic papers. My hesitation is being able to take that data and put it into a clinical context with the patient. So sometimes I look at who’s done the research and if there’s someone that I’m familiar with, who’s got a name in our industry with the experience clinically I am much more likely to take that on board as something that I can apply in my clinical setting, otherwise, I’m more skeptical about taking direct data out of the protocol and just putting that into my patient context.*
IG 3 P 3

## 4. Discussion

### 4.1. Principal Findings

Acupuncturists using a traditional Chinese medicine (TCM) framework spoke of how they saw themselves as a distinct practice that sat outside of mainstream medicine. This related to adapting pragmatic treatment approaches and seeking out education they saw as relevant. Acupuncture research was seen as being of limited value due to the common use of randomized controlled trials (RCTs) as the gold standard and its perceived inability to reflect clinical practice where an individualized diagnosis was crucial. However, practitioners also spoke of wanting to engage in research that would reflect clinical practice, suggesting case histories and observational data collection as possibilities. They also discussed how research would be more engaging if provided in formats such as summaries, peer group discussions, blogs, and lectures that related to clinical practice.

Practitioners in this study spoke of acupuncture research as being irrelevant to their clinical practice with the overwhelming critique that this was due to RCTs’ inability to use the individualized treatment they provided. Disease in TCM is viewed in terms of patterns of disharmony, where the body has lost its adaptive ability and this has caused an upset in the harmony, or equilibrium, present in the body. Disease is thus a manifestation of “a pattern of disharmonious relationships” within the body [26]. Therefore, it is not the disease itself that is being treated but the underlying imbalance, and by rearranging the pattern of disharmony into one of harmony and balance, the condition itself is treated [26]. The flexibility in differential diagnosis, including the continual reassessment of the “pattern” of disharmony, is considered one of the key features of TCM by practitioners [16] and is a component for assessing the quality of acupuncture clinical trials [27]. Another factor that may contribute to this view is that TCM practitioners have historically identified conflict between the theoretical frameworks of biomedicine and TCM, with TCM being “largely incompatible” with the mechanistic framework that underpins biomedicine [28] and, at least in New Zealand and Australia, practitioners often feel like they are not well-integrated into the mainstream healthcare system [29].

This perception from acupuncturists that research refers to irrelevant RCTs has recently been reported by Roberts and colleagues [29], but this seems to reflect a long standing objection to this form of clinical research [30]. The criticism that there is a disconnect between clinical research and real-life medical care of individual patients is also made within Western medicine, although rather than rejecting research, this discussion is framed around the need for different reporting and research strategies to inform clinical practice [31,32]. Surveys [33,34] and qualitative studies of acupuncture practitioners treating chronic health conditions, including lower back pain [35], depression [36], endometriosis [37], and infertility [38], provide support to the concept that treatment individuation is considered vital by practitioners. Despite this focus on differential diagnosis and subsequent individualization in clinical practice, there is little clinical research to support the superiority of this treatment approach [39] and there is some evidence to suggest that practitioners tend to use a core group of common points across a range of diagnoses rather than a diverse range of different points [40]. Crucially, several more recent acupuncture trials [10,12,41] in women’s health have used a technique called manualization [42] and a more pragmatic approach [42] to allow a significant degree of treatment individualization based on TCM differential diagnosis within the confines of the RCT.

It was noticeable that clinical research outside the scope of clinical trials was not mentioned by those interviewed in this study. Nonclinical trial publications could be used to reflect and build on clinical practice, create awareness for safety, and inform health professionals of their scope of practice. These include: an observational study on a hospital acupuncture maternity clinic in New Zealand [43] and a qualitative study of the importance of explanations and self-care advice in treating primary dysmenorrhea [17].

Practitioners saw case histories as being the most relevant to clinic practice. Case histories are acknowledged as having a role in medical education [44]. However, to be useful, they require a level of reporting that goes beyond a clinical anecdote, with pertinent detail and a specific focus to provide a clear rationale for why they contribute to current knowledge [44]. Although practitioners spoke of wanting to showcase what they saw as the benefits of acupuncture within their practice, the descriptive nature of a case history lacks the capacity to link cause and effect. However, it does allow for clinical observation, which can be valuable for practitioners [45], illustrating complexities of real-life practice and providing insights into the mind of the practitioner when diagnosing and treating [46]. With the availability of the CARE (for CAse REports) guidelines that include a specific checklist, a quality framework now exists for practitioners to use [47]. This offers guides for publishing case reports [48] or case series [49] in areas of integrative medicine such as acupuncture available for practitioners.

Practitioners reported, similar to our previous research [17], that they often could not treat as frequently as they liked due to their patients’ financial, time, and geographical constraints. Frequency and total number of treatments are an important part of the total ‘dose’ of acupuncture [9,50] both in research and in clinical practice. Given that both frequency and total number of treatments plays a role in acupuncture’s effectiveness for conditions such as depression [51], and the total number of treatments is a key factor in positive outcomes for acupuncture and IVF [13,52], failure to deliver enough treatment may mean that maximum therapeutic benefit is not achieved [53].

### 4.2. Strengths and Limitations

A limitation of this study was the limited variation and depth provided through the use of convenience/purposive sampling for the interview groups. Participants self-selected to be involved in this study, and of the 80 invited, only 11 eventually participated, with all practitioners having significant clinical experience. The demographic profiles of the interviewees do not represent an exact alignment with demographics of the professional acupuncture bodies from which they were recruited. Difficulties with recruitment of clinicians practicing women’s health who represent the underlying demographics of the profession in Australia and New Zealand have been reported by our team previously [17]. However, it is stated clearly throughout the analysis that only the viewpoints of those interviewed are being reported. In qualitative research, the experience and viewpoints of all those interviewed are of equal value, and while generalizability to wider populations is not the necessary intended goal in this type of analysis [29], it does not mean that the findings are not generalizable to experienced women’s health acupuncturists in Australia and New Zealand [54]. In addition, it must be noted that our findings are congruent with previous work undertaken in the mid 2000s on the role of research in informing clinical practice amongst both Australian acupuncture students and practitioners early in their career [15,16]. However, given the changing nature of the undergraduate curriculum, it is possible that more recent graduates may have different views on the role of research which were not captured in this study.

### 4.3. Implications for Clinical Practice and Research

There is a need, either during the undergraduate or post-graduate educational program itself or as part of ongoing professional education provided by professional organizations, to ensure that acupuncturists have the opportunity to engage in activities that promote a broad range of research literacy. This would support the ability to understand, critique, and subsequently feel confident as to when to implement research into their practice. For example, online journal clubs through professional bodies with a requirement for continuing educational development would provide a platform for informed discussion about the existing research base that can be utilized and build research literacy within the profession. Regular plain language summaries, similar to those provided by the Cochrane collaboration, may be useful and could be developed by experts in the area in conjunction with professional bodies. While meta-analyses and systematic reviews are still the pinnacle of the evidence pyramid, case reports and case series may be a more pragmatic entry point in research for practitioners. As a practical example, case reports or case series written up as per the CARE guidelines could form part of the curriculum in students’ final year. Clinical trial design for acupuncture is evolving to reflect concerns about the complex nature of TCM acupuncture [50], and this includes a move towards co-design, ensuring practitioners and clinicians are involved in the research design [55]. Alternatives to the traditional placebo-controlled trial design, some of which have previously been suggested as suitable for acupuncture research [50,56,57] including comparative effectiveness research [58], whole systems research [59], and n of 1 trials [60], should also be considered to help ensure that the complexity of community acupuncture practice is better reflected. This may address some of the concerns practitioners raised around the disconnect between clinical practice and clinical trials. Finally, the use of cost reducing measures such as group acupuncture clinics [61] may allow more frequent treatment for those who could otherwise not afford this, and thus allow practitioners to ensure that an adequate ‘dose’ of acupuncture is delivered.

## 5. Conclusions

Acupuncture practitioners treating women’s health conditions reported that they felt a disconnect between their clinical practice and the design of clinical trials, predominantly due to what they perceived as a lack of individualization of treatment, and therefore were often reluctant to incorporate research findings into their practice. Practitioners felt that case histories were the most valuable type of research, but these were not usually formally published in the academic literature. Improving research literacy and engagement is vital and this will require support from both educational institutions and professional bodies. Future clinical trials should also consider alternative designs that may increase their applicability to clinical practice.

## Figures and Tables

**Table 1 ijerph-18-00834-t001:** Interview group demographics.

Group	Interview Group 1 AUS + NZ	Interview Group 2 NZ	Interview Group 3 AUS
Gender			
Male	0	2	0
Female	3	2	4
Country			
Australia (AUS)	2	0	4
New Zealand (NZ)	1	4	0
Years in practice			
<5 years	0	0	0
5–10 years	0	0	0
>10 years	3	4	4

**Table 2 ijerph-18-00834-t002:** Overview of themes and subthemes.

Not Mainstream but a Stream	
Themes	Subthemes
Working with what you’ve got	Pragmatic treatment
	Working it out for yourself
Finding the right lens	Adapting research to a clinic lens
	More for the medics and patients than us
	Making research digestible
	Teachers you can trust

## Data Availability

The data presented in this study are available on request from the corresponding author. The data are not publicly available due to the conditions of the ethical approval.

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
