# Peer review of "The Role of Research in Guiding Treatment for Women’s Health: A Qualitative Study of Traditional Chinese Medicine Acupuncturists"

_ijerph, 2021, doi:10.3390/ijerph18020834_

Round 1
Reviewer 1 Report
It is interesting that authors conducted a qualitative study of TCM acupuncturists to examine the role of research in guiding treatment for women’s health, I think several concerns listed below will need to be addressed to improve paper quality
- The sample size ( n = 11) was too small to generalize the findings of this study.
- Please provide a full name for the abbreviation such as PMS, IVF, FG 2 P4
- Please describe the survey method in more detail.
- Please provide the ethical approval date.
Author Response
- The sample size ( n = 11) was too small to generalize the findings of this study.
Many qualtative studies of practitioners have sample sizes in this range and this does not limit generalisaiblity to certain groups per se. We have discussed this and provided context and references to support in our limitations section line 473-289
A limitation was the limited variation and depth provided through the use of convenience/purposive sampling for the interview groups . Participants self-selected to be involved in this study, and of the 80 invited only 11 eventually participated, with all practitioners having significant clinical experience. The demographic profiles of the interviewees do not represent an exact alignment with demographics of the professional acupuncture bodies from which they were recruited. Difficulties with recruitment of clinicians practicing women’s health who represent the underlying demographics of the profession in Australia and New Zealand have been reported by our team previously[18]. However, it is stated clearly throughout the analysis that only the viewpoints of those interviewed are being reported. In qualitative research the experience and viewpoints of all those interviewed are of equal value and while generalisability to wider populations is not the necessary intended goal in this type of analysis [30] it does not mean that the findings are not generalizable to experienced women’s health acupuncturists in Australia and New Zealand [56]. In addition it must be noted that our findings are congruent with previous work undertaken in the mid 2000s on the role of research in informing clinical practice amongst Australian both acupuncture students and practitioners early in their career [16,17]. However, given the changing nature of the undergraduate curriculum, it is possible that more recent graduates may have different views on the role of research which were not captured in this study.
- Please provide a full name for the abbreviation such as PMS, IVF, FG 2 P4
These have all been added and an explanation on the use of abbrevations for focus groups participants has been added to line 153-155
To ensure anonymity but to allow identification of different participants through various themes, each quotation indicated which interview group was used (IG) and which participant in that group was speaking (P), e.g. Participant 3 in Interview Group 1 would be represented as IG 1 P 3
- Please describe the survey method in more detail.
The survey itself will be published seperately as its a seperate component of the mixed methods project, however we have added the following details that outline the criteria for inclusion in both the survey and the interviews and also a guide to the content for the survey in line 91-101
The study population was drawn from the international survey (not yet published but whose content was similar to our previously published survey[6]) which had inclusion criteria of being a New Zealand and Australian acupuncturist, who were aged over 18, either a current member of Acupuncture NZ or New Zealand Acupuncture Standards Authority (NZASA) (if located in New Zealand) or hold current Chinese Medicine Practitioner registration with Australian Health Practitioner Regulation Agency (AHPRA) (if located in Australia). All small group interview participants were required to have been in clinical practice at least one day per week for the past 12 months and with a case load of at least 1/3 women’s health related conditions. Purposive sampling was used based on the assumption that those with significant experience in women’s health would yield significant information and provide unique perspectives based on their experience in the field [20] and in line with generally accepted approaches for qualitative descriptive analysis [21].
4. Please provide the ethical approval date.
Added line 87-88
Ethical approval was obtained from Western Sydney University Human Ethics Committee (H13099 – approved 5th February 2019).Reviewer 2 Report
Thank you for the opportunity to review this interesting research. The authors conducted a qualitative assessment regarding acupuncture research and its clinical implication to acupuncturists in Australia and New Zealand and found discrepancies between clinical practice and research in women’s health. The reviewer feels that the manuscript was coherent and well written, and for further improvement, the following comments would be useful for the authors.
1. the reviewer feels that the Table 1 is unnecessary because all information can be demonstrated in the manuscript.
#2. Some abbreviations are not well described, e.g. PMS, IVF, NZASA, AHPRA
#3. The authors may consider to discuss their system of acupuncture medicine in their society.
Author Response
1. the reviewer feels that the Table 1 is unnecessary because all information can be demonstrated in the manuscript.
We have altered Table 1 to provide more information, we would prefer to keep this as it provides the demographics and composition of each group at a glance.
#2. Some abbreviations are not well described, e.g. PMS, IVF, NZASA, AHPRA
We have added explanations for all abbreviations in the manuscript
#3. The authors may consider to discuss their system of acupuncture medicine in their society.
Thank you - we have added a short section in the discussion that mentions how practitioners position themselves in Australia and New Zealand.
Line 406-410
Reviewer 3 Report
Reviewer statement:
The Role of Research in Guiding Treatment for Women’s Health: A Qualitative Study of Traditional Chinese Medicine Acupuncturists
Title:
The title chosen reflect the study being reported and is considered adequate, reflecting on the study being performed.
Overall:
The paper is easy and pleasant to read, the English grammar is excellent.
Abstract:
The abstract is excellent.
Introduction
The introduction section is well written, stating the background of performing this study.
Methods
This section is well written and good to understand. Nevertheless, some points are missing or need clarification.
- The authors state on page 2, line 63: “The study population was drawn from New Zealand and Australian acupuncturists who were aged over 18, either a current member of Acupuncture NZ or NZASA (if located in New Zealand) or hold current Chinese Medicine Practitioner registration with AHPRA (if located in Australia)“. The abbreviations NZ, NZASA and AHPRA should be written out.
- The authors report that 15 participants were invited and only 11 participated. Could the authors provide the reason for the 4 participants not to participate?
- Furthermore, this should be considered a result, and should be reported in the appropriate section. Please do so.
Results
- From a reader perspective, the focus group consisted of only practitioners with > 10 years’ experience. This is very important as this can be an indication for bias. Please provide additional information to understand this point.
- Could this have influenced the reported results?
Discussion
This section was also pleasant to read.
- As mentioned earlier recruitment of only practitioners with > 10 years’ experience could have introduced bias, this should be acknowledge.
- The authors draw to my opinion the attention on the key point of the problem presented in the current article. On page 6 line 225: “ The criticism that there is a disconnect between clinical research and real life medical care of individual patients is also made within western medicine, although rather than rejecting research, this discussion is framed around the need for different reporting and research strategies to inform clinical practice [28,29]. Rather than criticizing of rejecting the method of research (RCT), evaluating and investigating the possibilities of different reporting and implementing strategies could be the aim for new research. This should be an invitation to participate in conducting studies, including RCT in which the underlying imbalance in patients can be establish, and thereafter randomization can be performed to establish which is the best treatment strategy. This message, in my opinion should be the key point of this article. Please elucidate on this point, and please marked this more clearly in the article.
Author Response
Thank you for your valuable feedback, our responses to the outstanding issues is documented below.
Methods
- The authors state on page 2, line 63: “The study population was drawn from New Zealand and Australian acupuncturists who were aged over 18, either a current member of Acupuncture NZ or NZASA (if located in New Zealand) or hold current Chinese Medicine Practitioner registration with AHPRA (if located in Australia)“. The abbreviations NZ, NZASA and AHPRA should be written out.
All abbrevations have now been added.
- The authors report that 15 participants were invited and only 11 participated. Could the authors provide the reason for the 4 participants not to participate?
- Furthermore, this should be considered a result, and should be reported in the appropriate section. Please do so
We have re-written the results and method section considerably and these are now addressed as follow on line 170-181 in the Results section
A total of 80 participants left their contact details indicating they meet the interview criteria and were interested in participating. Of these 48 were from Australia (AUS) and 32 from New Zealand (NZ). Fifteen returned the consent forms and agreed to participate with eleven participants finally taking part on the day - six from Australia and five from New Zealand. Of the four who signed consent forms and did not participate, all had last minute clinical commitments which prevented participation and could not make an alternate session. Three small interview groups were held with three to four participants in each group. Three practitioners were in a combined Australian and New Zealand group, four in an Australian group and four in a New Zealand group. All were members of their respective professional bodies, all held at least a Bachelors or equivalent level degree in acupuncture and ten of the 11 participants had done their undergraduate training in their respective country of current residence, with one participant having trained in China. Table 1 outlines the demographics of each interview group
Results
- From a reader perspective, the focus group consisted of only practitioners with > 10 years’ experience. This is very important as this can be an indication for bias. Please provide additional information to understand this point.
- Could this have influenced the reported results?
We have discussed the potential for bias and it's implications in more detail in the limitations section which has been re-written as follows on Line 473-489
A limitation was the limited variation and depth provided through the use of convenience/purposive sampling for the interview groups . Participants self-selected to be involved in this study, and of the 80 invited only 11 eventually participated, with all practitioners having significant clinical experience. The demographic profiles of the interviewees do not represent an exact alignment with demographics of the professional acupuncture bodies from which they were recruited. Difficulties with recruitment of clinicians practicing women’s health who represent the underlying demographics of the profession in Australia and New Zealand have been reported by our team previously[18]. However, it is stated clearly throughout the analysis that only the viewpoints of those interviewed are being reported. In qualitative research the experience and viewpoints of all those interviewed are of equal value and while generalisability to wider populations is not the necessary intended goal in this type of analysis [30] it does not mean that the findings are not generalizable to experienced women’s health acupuncturists in Australia and New Zealand [56]. In addition it must be noted that our findings are congruent with previous work undertaken in the mid 2000s on the role of research in informing clinical practice amongst Australian both acupuncture students and practitioners early in their career [16,17]. However, given the changing nature of the undergraduate curriculum, it is possible that more recent graduates may have different views on the role of research which were not captured in this study.
Discussion
This section was also pleasant to read.
- As mentioned earlier recruitment of only practitioners with > 10 years’ experience could have introduced bias, this should be acknowledge.
As noted above this is now dicussed in the limitations.
2. The authors draw to my opinion the attention on the key point of the problem presented in the current article. On page 6 line 225: “ The criticism that there is a disconnect between clinical research and real life medical care of individual patients is also made within western medicine, although rather than rejecting research, this discussion is framed around the need for different reporting and research strategies to inform clinical practice [28,29]. Rather than criticizing of rejecting the method of research (RCT), evaluating and investigating the possibilities of different reporting and implementing strategies could be the aim for new research. This should be an invitation to participate in conducting studies, including RCT in which the underlying imbalance in patients can be establish, and thereafter randomization can be performed to establish which is the best treatment strategy. This message, in my opinion should be the key point of this article. Please elucidate on this point, and please marked this more clearly in the article.
We have amended the implications for practice and research section to expand on the importance of looking at comparative effectiveness research and other trial designs that may overcome some of the shortcomings of using traditional placebo controls for complex interventions (Lines 606-614)
Clinical trial design for acupuncture is evolving to reflect concerns about the complex nature of TCM acupuncture [52], and this includes a move towards co-design, ensuring practitioners and clinicians are involved in the research design [57].Alternatives to the traditional placebo controlled trial design some of which have previously been suggested as suitable for acupuncture research [52,58,59], including comparative effectiveness research [60], whole systems research [61] and n of 1 trials [62] should also be considered to help ensure the complexity of community acupuncture practice is better reflected. This may address some of the concern’s practitioners raised around the disconnect between clinical practice and clinical trials.
Reviewer 4 Report
Overall, this is a very well written paper on an valuable topic. I have strong concerns around the methodology, particularly the claim that a reflexive thematic analysis (the term for Braun & Clarke’s TA) has been performed. I've selected 'major revision' because there are significant methodological matters to be addressed, but I suspect this could be rectified with a few thoughtful revisions. I wish the best of luck to the authors with these.
General comments
- NICM should be spelt out in the author affiliation and in the conflict of interest section.
- There is an overall tension in the writing of this paper that potentially reflects the assumption in our society that acupuncturists/CAM ‘should’ fit into western medicine/science. A sentence or two that succinctly acknowledges and critically explores this would enrich the paper.
Method
- A copy of the focus group interview guide as an online supplementary would increase the transparency of this work and aid the reader to assess the validity of the results.
- Page 2, line 78: How were the 15 potential participants selected and invited?
- The concept of saturation in qualitative research is heavily contested; see ‘Unsatisfactory Saturation’: a critical exploration of the notion of saturated sample sizes in qualitative research by O’Reilly & Parker 2012) for a nice summary. Further, Braun & Clarke have made it clear that it is no compatible with their approach to thematic analysis (which the authors have claimed to have used):
Reading such papers, we have discovered that we promote the use of codebooks and coding frames, consensus coding, the measurement of coding reliability, developing themes before data coding, data or theme saturation, the measurement and reporting of theme frequency, constant comparative analysis, and more . . . Reader, we do not! Not only are these things we have not said, they are all things we are indeed critical of, as practices for Big Q qualitative inquiry (Braun and Clarke 2013, 2019c; Clarke and Braun 2019). The most plausible (and perhaps generous) explanation for claims that we advocate for procedures that we do not in fact advocate for, is that the authors have not read our paper. (Source: One size fits all? What counts as quality practice in (reflexive) thematic analysis? by Braun & Clarke 2020)
- I see little evidence that Braun & Clarke’s TA has been performed. Data patterns are summarised but then no further interpretative analysis seeking to understand underlying patterns and ideologies appears; see the original Braun & Clarke TA paper and the one quoted above for further background. It would instead appear that a qualitative description has been conducted (see Whatever happened to qualitative description? by Sandelowski 2000), a perfectly legitimate analysis technique that is consistent with the study aim. If the authors want to continue to claim the use of Braun & Clarke’s TA, the results would been need to re-worked to demonstrate a latent analysis that goes beyond the data, as outlined in the original Braun & Clarke TA paper.
- A reflexive statement that outlines the positions, perceptions etc of the researchers in the methods—instead of the broad general statement in the strengths and limitations section—would be an asset for the reader’s interpretation of the findings. This need not be apologetic as we are all human and have our ‘biases,’ something that the qual methodology readily acknowledges and accepts.
- Page 7, line 268: ‘Interrater testing’ is also not consistent with Braun & Clarke’s thematic analysis, as stated in the quote above.
Results
- Is there further descriptive data on the participants, such as their education and training? It is difficult to assess the diversity of participants—a core quality assessment criteria in qualitative research—with the limited details provided.
- As stated above, the presented findings appear descriptive only. I also see little evidence of counter themes/cases, though this may reflect the lack of diversity in participants?
Discussion
- It is not the purpose of qualitative research to seek “exact alignment” with the population of interest. However, some scholars argue that we do seek generalisability through the use of diverse participants that capture as many aspects of phenomenon as possible. See https://journals.sagepub.com/doi/pdf/10.1177/104973299129121622 for a nice summary.
Author Response
Overall, this is a very well written paper on an valuable topic. I have strong concerns around the methodology, particularly the claim that a reflexive thematic analysis (the term for Braun & Clarke’s TA) has been performed. I've selected 'major revision' because there are significant methodological matters to be addressed, but I suspect this could be rectified with a few thoughtful revisions. I wish the best of luck to the authors with these.
Thank you, we appreciate both the insightful comments and the further reading provided, this was very helpful in helping the re-writing of the methods that was undertaken.
General comments
- NICM should be spelt out in the author affiliation and in the conflict of interest section.
Similar to IBM, NICM is not an abbreviation - our institute is NICM Health Research Institute. - There is an overall tension in the writing of this paper that potentially reflects the assumption in our society that acupuncturists/CAM ‘should’ fit into western medicine/science. A sentence or two that succinctly acknowledges and critically explores this would enrich the paper.
We agree and have added the following on Lines 406-410
Another factor that may contribute to this view is that TCM practitioners have historically identified conflict between the theoretical frameworks of biomedicine and TCM, with TCM being “largely incompatible” with the mechanistic framework that underpins biomedicine[29] and, at least in New Zealand and Australia, practitioners often feel like they are not well integrated into the mainstream healthcare system [30].
Method
- A copy of the focus group interview guide as an online supplementary would increase the transparency of this work and aid the reader to assess the validity of the results.
We have added the main questions in the Method (Lines 144-149) and uploaded the interview guide as a supplementary file. - Page 2, line 78: How were the 15 potential participants selected and invited?
We have re-written the methods section significantly - information on the selection is provided in lines 79-101
A qualitative approach using online small group interviews was seen as appropriate to seek out a deeper understanding of acupuncturists experiences and decision making. These interviews were part of a larger mixed methods study which included an international survey on how acupuncturists in Australia and New Zealand treated women’s health conditions. Survey participants who expressed a willingness to participate in the group interviews were sent an invitation to participate with information about how their data would be used and any comments used in the interviews deidentified. They were also given with several dates for the interviews in January and February 2020 so they could indicate their availability along with a consent from to return. Ethical approval was obtained from Western Sydney University Human Ethics Committee (H13099 – approved 5th February 2019). Small group interviews were chosen due to the barriers of getting larger numbers of practitioners together at one time due to clinical commitments.
The study population was drawn from the international survey (not yet published but whose content was similar to our previously published survey[6]) which had inclusion criteria of being a New Zealand and Australian acupuncturist, who were aged over 18, either a current member of Acupuncture NZ or New Zealand Acupuncture Standards Authority (NZASA) (if located in New Zealand) or hold current Chinese Medicine Practitioner registration with Australian Health Practitioner Regulation Agency (AHPRA) (if located in Australia). All small group interview participants were required to have been in clinical practice at least one day per week for the past 12 months and with a case load of at least 1/3 women’s health related conditions. Purposive sampling was used based on the assumption that those with significant experience in women’s health would yield significant information and provide unique perspectives based on their experience in the field [20] and in line with generally accepted approaches for qualitative descriptive analysis [21].
- The concept of saturation in qualitative research is heavily contested; see ‘Unsatisfactory Saturation’: a critical exploration of the notion of saturated sample sizes in qualitative research by O’Reilly & Parker 2012) for a nice summary. Further, Braun & Clarke have made it clear that it is no compatible with their approach to thematic analysis (which the authors have claimed to have used):
This has been removed from our re-written methodology as outlined below.
- I see little evidence that Braun & Clarke’s TA has been performed. Data patterns are summarised but then no further interpretative analysis seeking to understand underlying patterns and ideologies appears; see the original Braun & Clarke TA paper and the one quoted above for further background. It would instead appear that a qualitative description has been conducted (see Whatever happened to qualitative description? by Sandelowski 2000), a perfectly legitimate analysis technique that is consistent with the study aim. If the authors want to continue to claim the use of Braun & Clarke’s TA, the results would been need to re-worked to demonstrate a latent analysis that goes beyond the data, as outlined in the original Braun & Clarke TA paper.
While we did intend to undertake a Braun and Clarke TA we agree with the reviewer that in this case we did not hit the mark, potentially exacerbated to the way the questions were framed and the lack of diversity in our sample. We have re-written this as a qualitative descriptive study using a simpler version of TA. Lines 158-167 outline this
2.3 Data Analysis
A qualitative descriptive approach was used to formally analyse and code the data[23], Qualitative description research seeks instead to provide a rich description of the experience depicted in easily understood language [24] and seek to discover and understand a phenomenon, a process, or the perspectives and worldviews of the people involved [25]. A thematic analysis was undertaken as per Boyatzis [26]. Initial coding was carried out by one researcher (SA) and checked and after feedback and discussion verified by DB.Dominant and/or repeated codes were then categorized into themes along with quotations summarizing specific themes. After a consensus was reached regarding classification of codes and themes, quotations in each domain were summarized and presented in a qualitative descriptive manner.
- A reflexive statement that outlines the positions, perceptions etc of the researchers in the methods—instead of the broad general statement in the strengths and limitations section—would be an asset for the reader’s interpretation of the findings. This need not be apologetic as we are all human and have our ‘biases,’ something that the qual methodology readily acknowledges and accepts.
We have added this in Line 103-108
2.1 Positioning of the Researcher
All of the researchers have formal acupuncture qualifications with three of these (MA, DB, KR) remaining involved in providing acupuncture in a clinical practice setting. With previous publications involving acupuncture use women’s health (MA, DB & CS) it is important to recognize as researchers this confers an ‘insiders perspective’ and specific interest in designing and implementing research to explore how research translates into clinical practice for acupuncturists.
- Page 7, line 268: ‘Interrater testing’ is also not consistent with Braun & Clarke’s thematic analysis, as stated in the quote above.
This has been removed.
Results
- Is there further descriptive data on the participants, such as their education and training? It is difficult to assess the diversity of participants—a core quality assessment criteria in qualitative research—with the limited details provided.
We have added extra information on the participants on line 170-180
A total of 80 participants left their contact details indicating they meet the interview criteria and were interested in participating. Of these 48 were from Australia (AUS) and 32 from New Zealand (NZ). Fifteen returned the consent forms and agreed to participate with eleven participants finally taking part on the day - six from Australia and five from New Zealand. Of the four who signed consent forms and did not participate, all had last minute clinical commitments which prevented participation and could not make an alternate session. Three small interview groups were held with three to four participants in each group. Three practitioners were in a combined Australian and New Zealand group, four in an Australian group and four in a New Zealand group. All were members of their respective professional bodies, all held at least a Bachelors or equivalent level degree in acupuncture and ten of the 11 participants had done their undergraduate training in their respective country of current residence, with one participant having trained in China - As stated above, the presented findings appear descriptive only. I also see little evidence of counter themes/cases, though this may reflect the lack of diversity in participants?
We believe the above alterations to the method will have resolved this concern.
Discussion
- It is not the purpose of qualitative research to seek “exact alignment” with the population of interest. However, some scholars argue that we do seek generalisability through the use of diverse participants that capture as many aspects of phenomenon as possible. See https://journals.sagepub.com/doi/pdf/10.1177/104973299129121622 for a nice summary.
We have expanded on this in our limitations section (Line 473-489)
A limitation was the limited variation and depth provided through the use of convenience/purposive sampling for the interview groups . Participants self-selected to be involved in this study, and of the 80 invited only 11 eventually participated, with all practitioners having significant clinical experience. The demographic profiles of the interviewees do not represent an exact alignment with demographics of the professional acupuncture bodies from which they were recruited. Difficulties with recruitment of clinicians practicing women’s health who represent the underlying demographics of the profession in Australia and New Zealand have been reported by our team previously[18]. However, it is stated clearly throughout the analysis that only the viewpoints of those interviewed are being reported. In qualitative research the experience and viewpoints of all those interviewed are of equal value and while generalisability to wider populations is not the necessary intended goal in this type of analysis [30] it does not mean that the findings are not generalizable to experienced women’s health acupuncturists in Australia and New Zealand [56]. In addition it must be noted that our findings are congruent with previous work undertaken in the mid 2000s on the role of research in informing clinical practice amongst Australian both acupuncture students and practitioners early in their career [16,17]. However, given the changing nature of the undergraduate curriculum, it is possible that more recent graduates may have different views on the role of research which were not captured in this study.
Round 2
Reviewer 1 Report
Thank you for your reply